# End-to-End Learning to Follow Language Instructions with Compositional Policies

**Vanya Cohen**†
The University of Texas at Austin
vanya@utexas.edu

**Geraud Nangue Tasse**†
University of the Witwatersrand
geraudnt@gmail.com

**Nakul Gopalan**
Georgia Institute of Technology
nakul_gopalan@gatech.edu

**Steven James**
University of the Witwatersrand
steven.james@wits.ac.za

**Raymond Mooney**
The University of Texas at Austin
mooney@utexas.edu

**Benjamin Rosman**
University of the Witwatersrand
benjamin.rosman1@wits.ac.za

## Abstract

We develop an end-to-end model for learning to follow language instructions with compositional policies. Our model combines large language models with pretrained compositional value functions [Nangue Tasse et al., 2020] to generate policies for goal-reaching tasks specified in natural language. We evaluate our method in the BabyAI [Chevalier-Boisvert et al., 2019] environment and demonstrate compositional generalization to novel combinations of task attributes. Notably our method generalizes to held-out combinations of attributes, and in some cases can accomplish those tasks with no additional learning samples.

## 1  Introduction

This work extends the paper "Learning to Follow Language Instructions with Compositional Policies" Cohen et al. [2021] to enable end-to-end, compositional learning of language-instruction following tasks. We build on the Boolean-compositional value function representations of Nangue Tasse et al. [2020] and propose an end-to-end system for learning compositional policies for following language instructions. By leveraging the few-shot learning properties of the large language model T5 Raffel et al. [2020], in combination with compositional value functions, we aim to demonstrate more sample efficient learning of goal-reaching tasks specified by language commands. We evaluate learning to follow language instructions in the BabyAI domain Chevalier-Boisvert et al. [2019].

In Cohen et al. [2021] the value functions and instruction-to-Boolean-expression translation model are first learned separately and then combined during inference. The compositional value functions were trained using the procedure outlined in Nangue Tasse et al. [2020] and the translation model

---

† Equal contribution

36th Conference on Neural Information Processing Systems (NeurIPS 2022).

was learned using reinforcement learning. In this work, we instead learn compositions in an end-to-end manner. We draw inspiration from past work in Neural Module Networks (NMN) Andreas et al. [2015] Hu et al. [2018]. These networks learn an input-conditioned layout of differentiable modules that reflect the compositional structure of each task. Our proposed Neural Module Value Network (NMVN) implements a differentiable Boolean composition of pretrained compositional value functions modules. The model uses pretrained language representations from T5. The NMVN optimizes the same loss function used to train the compositional value functions.

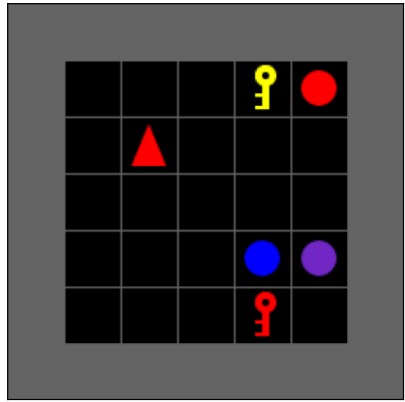

Figure 1: Example of a task in the BabyAI domain. Here the agent (red triangle) needs to complete the mission specified by the command "pick up the red key". Solving this task with compositional value functions requires using the conjunction of the *pickup* "red object" and "key" value functions.

Previous work has demonstrated the challenges of systematic compositional generalization in neural networks and proposes that a failure to learn compositional representations may be responsible for the sample inefficiency of deep learning methods [Lake and Baroni, 2018]. Many language instruction tasks posses compositional structure. The meaning of a compositional command can be determined from the meaning of component phrases and their arrangement [Szabó, 2020]. While deep learning methods have attained success in language instruction following tasks [Blukis et al., 2019] [Chaplot et al., 2018] [Tambwekar et al., 2021], these methods require large amounts of training data and do not generalize to novel compositions of instruction components. By utilizing the few-shot generalization capacity of large language models [Brown et al., 2020] in combination with the sample efficiency of Boolean compositional policies [Nangue Tasse et al., 2020] we build a system which can generalize compositionally to novel tasks without requiring large amounts of additional training data.

We present sample-efficiency results for learning in the BabyAI domain that show learning novel compositions of task attributes requires an order of magnitude fewer environment samples than the model of Cohen et al. [2021]. These experiments test the ability of the agent to generalize both to novel compositions of goal conditions and entirely unseen goal conditions. The proposed method is evaluated against a non-compositional baseline based on the deep Q-network (DQN) BabyAI model [Mnih et al., 2015]. Further we show that the network learns to properly utilize individual value function modules in a compositional manner.

This project makes the following contributions:

- We propose a novel model, which learns in an end-to-end manner to connect pretrained compositional value functions with language representations. The model directly maps language and pixel-level environment observations to composed value functions that determine policies for acting in the environment and learns from the same Q-learning loss used to train the DQN-based [Mnih et al., 2015] compositional value functions.

- We show learning results for three task curricula designed to test compositional learning in BabyAI and show that our model leads to substantial savings in the number of samples required to learn novel compositions of tasks over our original model in Cohen et al. [2021] and a non-compositional baseline.

- We find that the internal representations learned by our model match the underlying compositional task structures.

- We detail the challenges and advantages of learning language representations from a value-function-approximation learning signal. This signal differs from the majority of other weakly supervised language learning objectives, including our original model in Cohen et al. [2021], which learns using policy-gradient methods.

## 2 Background

We consider the case of an agent required to solve a series of related tasks. Each task is formalized as a Markov decision process (MDP) $\langle \mathcal{S}, \mathcal{A}, p, r \rangle$, where $\mathcal{S}$ is the state space and $\mathcal{A}$ is the set of actions available to the agent. The transition dynamics $p(s'|s, a)$ specify the probability of the agent entering state $s'$ after executing action $a$ in state $s$, while $r(s, a, s')$ is the reward for executing $a$ in $s$. We further assume that $r$ is bounded by $[r_{\text{MIN}}, r_{\text{MAX}}]$. We focus here on goal-reaching tasks, where an agent is required to reach a set of terminal goal states $\mathcal{G} \subseteq \mathcal{S}$.

In our formulation, tasks are related in that they differ only in their reward functions. Specifically, we first define a background MDP $M_0 = \langle \mathcal{S}_0, \mathcal{A}_0, p_0, r_0 \rangle$. Then, any new task $\tau$ is characterized by a task-specific reward function $r_\tau$ that is non-zero only for transitions entering $g$ in $\mathcal{G}$. Consequently, the reward function for the resulting MDP is given by $r_0 + r_\tau$.

The agent aims to learn an optimal policy $\pi$, which specifies the probability of executing an action in a given state. The value function of policy $\pi$ is given by $V^\pi(s) = \mathbb{E}_\pi \left[ \sum_{t=0}^\infty r(s_t, a_t) \right]$ and represents the expected return after executing $\pi$ from $s$. Given this, the optimal policy $\pi^*$ is that which obtains the greatest expected return at each state: $V^{\pi^*}(s) = V^*(s) = \max_\pi V^\pi(s)$ for all $s \in \mathcal{S}$. Closely related is the action-value function, $Q^\pi(s, a)$, which represents the expected return obtained by executing $a$ from $s$, and thereafter following $\pi$. Similarly, the optimal action-value function is given by $Q^*(s, a) = \max_\pi Q^\pi(s, a)$ for all $(s, a) \in \mathcal{S} \times \mathcal{A}$.

### 2.1 Logical Composition of Tasks and Value Functions

Recent work [Nangue Tasse et al., 2020] has demonstrated how logical operators such as conjunction ($\wedge$), disjunction ($\vee$) and negation ($\neg$) can be applied to value functions to solve semantically meaningful tasks with no further learning. To achieve this, the reward function is extended to penalise the agent for attaining goals it did not intend to:

$$\bar{r}(s, g, a) = \begin{cases} \bar{r}_{MIN} & \text{if } g \neq s \in \mathcal{G} \\ r(s, a) & \text{otherwise,} \end{cases} \tag{1}$$

where $\bar{r}_{MIN}$ is a large negative penalty. Given $\bar{r}$, the related value function, termed *world value function* [Nangue Tasse et al., 2022], can be written as

$$\bar{Q}(s, g, a) = \bar{r}(s, g, a) + \int_{\mathcal{S}} \bar{V}^{\bar{\pi}}(s', g) p(s'|s, a) ds', \tag{2}$$

where $\bar{V}^{\bar{\pi}}(s, g) = \mathbb{E}_{\bar{\pi}} \left[ \sum_{t=0}^\infty \bar{r}(s_t, g, a_t) \right]$.

These value functions are intrinsically *compositional* since if a task can be written as the logical expression of previous tasks, then the optimal value function can be derived by composing the learned world value functions similarly. For example, consider the `PickUpObj` domain shown in Figure 1. Imagine that the agent has separately learned the task of collecting red objects (task $R$) and keys (task $K$). Using these value functions, the agent can immediately solve the tasks defined by their union ($R \vee K$), intersection ($R \wedge K$), and negation ($\neg R$) as follows:

$$\bar{Q}^*_{R \vee K} = \bar{Q}^*_R \vee \bar{Q}^*_K := \max\{\bar{Q}^*_R, \bar{Q}^*_K\}$$

$$\bar{Q}^*_{R \wedge K} = \bar{Q}^*_R \wedge \bar{Q}^*_K := \min\{\bar{Q}^*_R, \bar{Q}^*_K\}$$

$$\bar{Q}^*_{\neg R} = \neg \bar{Q}^*_R := (\bar{Q}^*_{MAX} + \bar{Q}^*_{MIN}) - \bar{Q}^*_R,$$

where $\bar{Q}^*_{MAX}$ and $\bar{Q}^*_{MIN}$ are the world value functions for the *maximum* and *minimum* tasks respectively.[1]

---

[1]The maximum task is defined by the reward function $r = r_{\text{MAX}}$ for all $\mathcal{G}$. Similarly, the minimum task has reward function $r = r_{\text{MIN}}$ for all $\mathcal{G}$.

## 2.2 T5 Pretrained Transformer Language Representations

Transformer language models use the self-attention mechanism Vaswani et al. [2017] to generate abstract sequence representations of text inputs and transform these representations into probability distributions over text outputs. Following the original Transformer architecture, the T5 model contains both a text encoder and decoder. Each of these is a stack of self-attention layers that transform input sequences to output sequences 2.

We use the T5 sequence-to-sequence model Raffel et al. [2020] based on the Transformer architecture Vaswani et al. [2017] to generate sentence embeddings of the input BabyAI mission commands. The large-scale pretraining scheme of T5 and subsequent fine-tuning on downstream tasks, has significantly improved performance on diverse NLP tasks such as classification, translation, and question answering [Devlin et al., 2018, Peters et al., 2018, Radford et al., 2018, 2019] T5 is pretrained on the Colossal Clean Crawled Corpus (C4) Raffel et al. [2020], a filtered version of the Common Crawl.[2]. During pretraining, the model optimizes a self-supervised reconstruction objective to learn semantic representations of the language corpus. We chose T5 because Raffel et al. perform numerous ablations to develop their pretrained models and demonstrate high performance on a variety of NLP tasks.

## 3 Related Work

Our work is situated within the paradigm of reinforcement learning where novel tasks are specified using natural language and the agent is required to solve the task in the fewest possible steps. Previous approaches have solved this problem using end-to-end architectures that are learned or improved using reinforcement learning Anderson et al. [2018], Blukis et al. [2019], Chaplot et al. [2018]. A problem with such approaches is a lack of compositionality in the learned representations. Specifically, learning to navigate to a red ball does not help the agent to learn to identify and navigate to a blue ball. Approaches that translated language commands to a symbolic representation and then planned to get to the goal can demonstrate compositionality due to the pre-specified symbolic representations Dzifcak et al. [2009], Williams et al. [2018], Gopalan et al. [2018]. However these works do not allow the agent to learn policies, but use pre-specified symbols and a model for planning. Compositional representation learning has been demonstrated in the solving computer vision and language processing tasks using Neural Module Networks (NMN) Andreas et al. [2015] Hu et al. [2018], but we explicitly want to learn a compositional representation both for the reinforcement learning policies and the language command. Kuo et al. [2021] do demonstrate compositional representations for policy, but they depend on a pre-trained parser to learn this representation. On the other hand we use large language models [Raffel et al., 2020] and compositional policy representations to demonstrate compositionality in our representations and the ability to solve novel unseen instruction combinations.

Compositional policy representations have been demonstrated using value function compositions, which were first demonstrated by Todorov [2007] using the linearly solvable MDP framework. Moreover, zero-shot disjunction [Van Niekerk et al., 2019] and approximate conjunction [Haarnoja et al., 2018a, Van Niekerk et al., 2019, Hunt et al., 2019] have been shown using compositional value functions. Nangue Tasse et al. [2020] demonstrate zero-shot optimal composition for all three logical operators—disjunction, conjunction, and negation—in the stochastic shortest path problems. Our approach extends ideas from Nangue Tasse et al. [2020] to solve novel commands specified using natural language.

## 4 Methods

### 4.1 Learning the Compositional Value Functions

Like Nangue Tasse et al. [2020], we use deep Q-learning Mnih et al. [2015] to learn the Q-function for each goal of the compositional value functions. We represent each compositional value function $\bar{Q}^*$ with a list of $|\mathcal{G}|$ DQNs, such that the Q-function for each goal $Q_g^*(s, a) \coloneqq \bar{Q}^*(s, g, a)$ is approximated with a separate DQN.

---

[2]https://commoncrawl.org

For each task, the agent starts training after 1000 steps of random exploration to populate an experience replay buffer and a goal buffer (set of reached terminal states). For each episode, the agent samples a random goal from the goal buffer and uses $\epsilon$-greedy to act in the environment. For each action, $a$, that the agent takes in each state, $s$, it receives goal-oriented rewards (Equation 1) given by:

$$\bar{r}(s, g, a) = \begin{cases} -0.1 & \text{if } g \neq s \in \mathcal{G} \\ r(s, a) & \text{otherwise,} \end{cases}$$

where task reward $r(s, a) = 2$ for picking up the correct object and $r(s, a) = -0.1$ everywhere else.[3] The episode terminates after the agent picks up any object. The agent's compositional value function is then trained per episode using the collected experience. Training ends once the agent reaches a success rate of at least $0.98$. For lower success rates, the compounding effect of composing sub-optimal policies negatively impacts the translation model's learning.

## 4.2 Neural Module Value Network

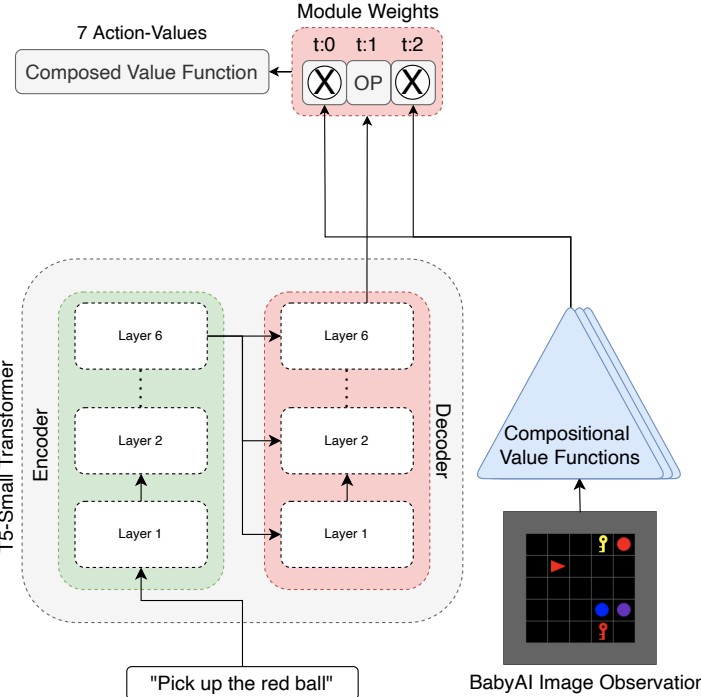

Figure 2: The NMVN architecture used to learn compositional value functions. The network maps image observation inputs and text BabyAI missions to Q-values by composing the pretrained compositional value functions using a differentiable attention mechanism. This model learns using the same DQN objective that was used to train the compositional value functions.

The Neural Module Value Network in Figure 2 maps input BabyAI observations and text mission statements to Q-values. The BabyAI environment observations are image observations of the whole environment and BabyAI mission commands. The image observations are 54 by 54 pixels and contain 3 color channels for red, green, and blue. The commands take the form "pick up [the/a] [object]" where [object] contains a composition of type and color attributes (e.g. red box). If more than one valid goal object is present in the environment, the indefinite article "a" is used.

In the case of the NMVN, T5 is used to produce a sequence of embeddings of a fixed output length conditioned on the input mission command. This usage shares some similarities with translation tasks, where one text sequence is mapped to another in an end-to-end manner. However in the NMVN the output representations are not decoded to a Boolean expression text sequence. Instead they are transformed into attention values over the pretrained compositional value functions and operations.

---

[3]We used $\bar{r}_{MIN} = r_{MIN} = -0.1$ since that is the simplest choice and it did not result in any discernible change in the success rate of the composed policies.

As there are two object attribute classes available in BabyAI (color and object type) the model allows for Boolean conjunction expressions with two arguments. At each argument position of the decoder sub-module, attention weights are calculated over a vocabulary of tokens describing the object type attributes $\{box, ball, key\}$, object colors $\{red, blue, green, grey, purple, yellow\}$. At the operator position, attentions are calculated over the two logical operators $\{and, or\}$ however *or* attentions are not used as the tasks examined do not require disjunction. Attentions are calculated as the softmax over these token logits. We also add an additional value function for going to any object represented by the token $\{object\}$. While this token is present during training it is not required to solve the environments evaluated in this project. To approximate the min and max operations from Nangue Tasse et al. [2020], we utilize softmin and softmax with a temperature setting.

At each argument position, a hard attention is calculated from the soft-attention described by taking the max over attention positions. As such the attention mechanism learns to select the appropriate arguments to the conjunction expression. However by utilizing a hard attention mechanism the model is no longer end-to-end differentiable without a further modification. We approximate the gradient with respect to the hard attention using the straight-through estimator of Jang et al. [2016]. In the forward pass all argument attentions are hard, but the gradients for the backward pass are calculated with respect to a softmax distribution. We find this works well in practice and produces policies which attain our success threshold of 95%, which was not true for the soft attention-based arguments.

A soft attention mechanism is used to select the appropriate Boolean operator from $\{and, or\}$. At each position both operators are evaluated with respect to the input arguments; however, in these experiments we only evaluate with respect to the *and* operator output. Composed value function outputs are propagated in a left-right manner using a simple recurrence mechanism. As with the first argument on the left-hand side of the Boolean composition at t0, the output of the composition can be passed as the left-hand side argument to further Boolean operations. We note that this restricts the space of compositions expressible by the model and some tasks would require additional attention and memory modules to handle operation precedence. Nonetheless this model is sufficient to express the compositions needed to solve the BabyAI navigation tasks.

### 4.3 Learning to Compose Value Functions

In preliminary experiments that utilized soft attention mechanisms over arguments, we found that the model would get stuck in a local maximum of averaging the attention across the appropriate value functions. For example in the task "pick up the red ball" the model would place half of its attention for each argument on the value functions for *red* and *ball*. These averaged value function arguments resulted in relatively high performing, but not optimal, policies. Modifications to the reward function to eliminate these local maxima yielded significantly degraded performance, likely because the reward scale from the environment no longer matched that of that of the compositional value functions.

The optimization of the NMVN differs significantly from that of Cohen et al. [2021] and posed numerous challenges, while also conferring some benefits. The original model also utilizes pretrained value functions, but learns to translate mission commands to output text Boolean expressions. These text outputs are then parsed and used to instantiate composed value functions to determine policies to act in the environment. While this translation model was also trained using reinforcement learning from environment rewards, it was optimized using a policy-gradient loss. The reward signal was derived from the average reward from 50 policy rollouts for each sampled environment. Prior work has established that policy-gradient methods suffer from high variance and require variance reduction techniques (in this case averaging the policy rollouts) to work well in practice Greensmith et al. [2004].

The NMVN instead learns from a deep Q-learning, temporal difference (TD) loss Mnih et al. [2015], Sutton [1988]. Instead of maximizing the expected reward of the policy rollout in the environment, the NMVN agent instead minimizes the temporal difference error in estimating the Q-values at each image-state, where the space of possible Q-values are provided by the pretrained compositional value functions. The model is capable of expressing a wide number of potential value functions, due to the number of combinations of compositional value function arguments. Learning to minimize this TD error (when leveraging pretrained value functions) should require fewer interactions with the environment, as every step in the environment provides a lower-variance value function Greensmith

et al. [2004] than the higher-variance policy rollout estimates. Our experiments show the new end-to-end agent learns with fewer environment interactions than the previous agent.

Although the learning signal may be lower variance, optimizing DQN poses significant challenges and issues with stable learning are outlined in other works Haarnoja et al. [2018b], Maei et al. [2009], Mnih et al. [2015], van Hasselt et al. [2015]. Through a rigorous hyperparameter search we find the NMVN DQN learning algorithm sensitive to small hyperparameter changes. For example, when calculating the value function attentions in the straight-through estimator, the algorithm requires a particular setting for the softmax temperature Ackley et al. [1985] and otherwise does not converge. Further we found learning sensitive to the noise introduced by dropout in the language model and consequently turned off dropout. Lastly, performance is sensitive to the learning rate used. In these experiments we settled on a significantly smaller learning rate than is generally used with our optimizer, but found it still produced fast and correct learning. Carefully tuning the off-policy exploration variable $\epsilon$ also resulted in significant improvements to agent learning. A significant fraction of the time spent on experiments went into identifying the various issues caused by slight variations in these parameter settings, and required tuning the parameters in a fully-supervised pseudo environment to identify reasonable settings. A full list of relevant hyperparameters are available in the Appendix.

## 5 Results

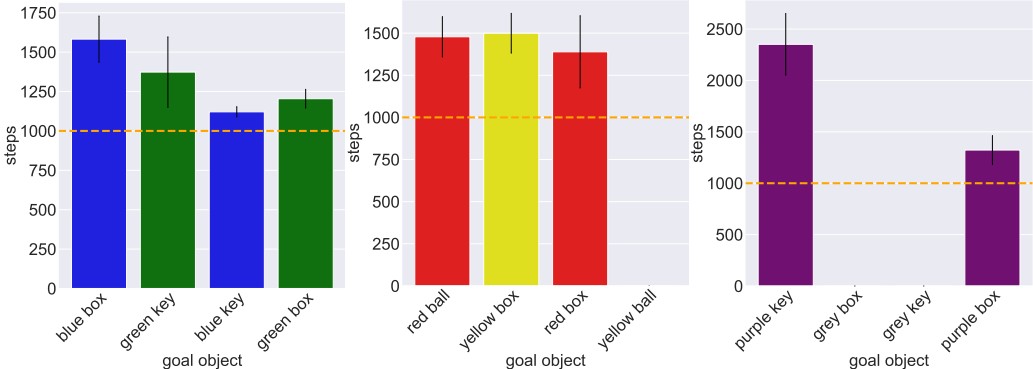

(a) Task set for the attributes $\{box, key, green, blue\}$. The agent requires significantly fewer learning steps to learn each additional task. The agent requires fewer steps to learn the held-out combinations of attributes. In both cases the agent requires at most approximately 250 environment steps.

(b) Task set for the attributes $\{box, ball, yellow, red\}$. The agent requires no additional steps to learn the held-out "pick up the yellow ball" task but roughly the same number of steps to learn the "pick up the red box" task.

(c) Task set for the attributes $\{key, box, purple, grey\}$. In this sequence the middle two tasks now share color attributes. The agent requires no additional steps to learn the held-out "pick up the grey box" and "pick up the grey key" tasks and requires fewer steps to learn the "pick up the purple box" task.

Figure 3: Results for the NMVN. The pickup task is learned for each object in series. For three trials, the mean environment steps needed to attain a 95% success rate are plotted for each task. Note that some tasks require no additional learning for the agent to succeed. Standard deviations are indicated for all tasks. During the first 1,000 steps no learning takes place as the DQN replay buffer is filled. The dashed line indicates the number of steps after which agent learning starts.

We show learning results for sequences of tasks that test the agent's ability to generalize compositionally to novel compositions of attributes and novel attributes entirely. Figure 3 shows results for the three task sequences investigated. The NMVN is trained using a warm-up period of 1,000 environment steps to fill the experience replay buffer for DQN training. Even though learning only starts after this period, we include these steps in the results for comparison purposes. Importantly if the agent cannot generalize immediately to the novel task, the learning steps required to solve the task starts at 1,000. In each task the agent needs to pick up a correct goal object, where four "distractor" objects are sampled uniformly at random from all possible object types. There may be more than one valid goal object in the environment and this changes the language command to refer to the object

using the indefinite article "a." Evaluation takes place at the start of learning for each new task, and after every 10 episodes. The evaluation tests the agent in 100 randomized task environments. To pass the evaluation and start learning the next task, the agent needs to attain a 95% success rate.

The plots in Figure 3 test different combinations of object attributes in varying sequences. Compositional generalization would mean that tasks defined by novel combinations of learned object attributes require significantly fewer learning steps than the original tasks. Indeed for all sequences investigated, the two novel combinations require fewer learning steps than the first two base tasks. Notably, the model does not require additional learning steps to solve some tasks, and instead relies upon transfer learning from the pretrained language space and task-specific fine-tuning to solve these environments. In Figure 3a and 3c the final task requires more learning steps to solve than the penultimate task. If this is evidence of overfitting, it may be possible to ameliorate with better hyperparameter choices and regularization. Differences in sample complexity between tasks are likely the result of prior probabilities over the attentions for those value function arguments.

Investigations of the attentions learned across all NMVN trials and tasks indicate that the agent learns the correct attentions for the value functions when solving each task. Evaluations of agents with nearly correct attentions (e.g. one argument correct) confirmed that the agent cannot attain a high success rate unless it selects the correct value function compositions for each task. Altogether, these findings imply that the NMVN learns more efficiently then the original agent in Cohen et al. [2021] without sacrificing model interpretability which is one of the main motivations behind the use of NMN Andreas et al. [2015].

While the NMVN agent and the agent from Cohen et al. [2021] require roughly the same number of optimization steps to learn the tasks presented, the agent in Cohen et al. [2021] requires more than an order of magnitude more environment interactions in the form of policy rollouts to attain a usable learning signal. The NMVN requires significantly fewer environment interactions to learn than the non-end-to-end model and trains more quickly as a result. This indicates that the use of end-to-end optimization in combination with a new objective dramatically improves sample efficiency for learning to compose the value functions for BabyAI tasks.

In contrast to our method, the BabyAI baseline is a joint language and vision model which learns a single Q-function from scratch for all tasks. The baseline architecture is based on a CNN-DQN Mnih et al. [2015] with a GRU Cho et al. [2014] implementing the BabyAI mission encoder. The final GRU hidden state is used as the representation for the input mission command. Like the NMVN this network maps image observation inputs and text BabyAI missions to Q-values, but utilizes a FiLM layer Perez et al. [2017] to condition the Q-values on the inputs. Like the pretrained compositional value functions and NMVN, this model also learns using the DQN objective. Its component value function and language representations are not pretrained. Both the NMVN and previous model learn both task value functions and language separately and then learn to combine them compositionally. The baseline model does not have explicit compositional representation. Although not a head-to-head comparison, we also ran evaluations for the baseline agent on these curricula, and found that it did not successfully learn any of the tasks in fewer than 15,000 steps. This suggests that the model is not able to generalize effectively between tasks.

## 6  Conclusion and Future Work

We validate that an end-to-end approach is capable of learning to efficiently compose pretrained value functions. Given these confirmatory experiments, future work can investigate the changes required to simultaneously optimize compositional value functions and the model. Additionally, because of the challenge of getting the end-to-end model to train, there may be additional performance that can be obtained through further hyperparameter and architecture search. Another potential axis of compositional generalization involves learning to generalize between Boolean operators. These experiments would require modifications to the BabyAI domain to support tasks that can utilize both disjunctive and conjunctive task semantics. Further work in expanding the BabyAI domain can also enable experiments to train a modified NMVN with mechanisms to handle operator precedence for specifying arbitrary combinations of task attributes. Lastly as discussed in the introduction, extensions to experiments in Habitat could test new methods for learning compositional value functions in more challenging and realistic environments. These environments also open up possibilities for more complex language commands and compositional task structures.

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

# A  Appendix

| NMVN Hyperparameters | |
|---|---|
| Dropout | 0.0 |
| Optimizer | AdamW |
| Learning rate | 1e-6 |
| Softmax Temp | 0.5 |
| Replay Buffer Size | 1e3 |
| $\epsilon$ init | 0.5 |
| $\epsilon$ final | 0.1 |

Table 1: The model hyperparameters were determined empirically through grid-search over a set of held-out tasks. The AdamW optimizer was introduced by Loshchilov and Hutter [2019].

