# OpenReview forum: "End-to-End Learning to Follow Language Instructions with Compositional Policies"
_robot-learning.org/CoRL/2022/Workshop/LangRob — LangRob 2022 Poster_

### Official Review · Reviewer_RcPJ · 2022-11-11
**Good paper, relevant to the conference.**

**Rating:** 8
**Confidence:** 4

**Review:**

This paper proposes a method that takes inspiration from neural module networks to compose previously-learned value functions and solve novel tasks specified in language, whereby the novel tasks are compositions in first-order logic over the previously learned tasks.
The proposed architecture uses attention mechanisms to learn how to apply the boolean relations to combine the apriori learned value functions. The paper performs experiments on the BabyAI dataset, showing that the model adapts to compositional combinations of known tasks much faster than it takes to learn completely novel tasks.

Pros:
- Very interesting way of combining ideas from neural module networks with value function learning.
- Good results.
- Relevant to the workshop topic.

Cons:
- None that are not already intended in the future work.

Minor Suggestions:
- Some sentences are a little hard to parse, for example: "Our proposed Neural Module Value Network (NMVN) utilizes pretrained language representations from T5 to express a differentiable Boolean composition of pretrained compositional value functions modules"
- Line 35: It's true that the cited methods require a lot of training data, but empirically they do generalize to novel compositions of instruction components.

---

### Official Review · Reviewer_VGBD · 2022-11-13
**Language conditioned composition of value functions.**

**Rating:** 7
**Confidence:** 4

**Review:**

The paper proposes a method that uses pretrained language representations from T5 to compose pretrained compositional value function modules using logical operators. The proposed model uses the attention mechanism over the pretrained value functions and operations to synthesize a composition of the same.

Strengths
1. Interesting results on out-of-distribution bi-gram compositions.
2. De-couples learning individual value functions from the procedure of combining them resulting in sample efficiency gains.

Weaknesses
1. A discussion of what kind of value functions are and are not expressable using such a model would be interesting.
2. The title is a bit misleading. There is no end-to-end learning of the compositional policies. The value functions are learned in advance of the "composing" procedure.
3. Implementation details are coupled within the text describing the idea, which results in the paper being hard to read.
4. The efficacy of the proposed method beyond the bi-gram setting of the problem is unclear.

---

### Decision · Program_Chairs · 2022-11-15

Accept (Poster)